# Characterization of 3D Organotypic Culture of Mouse Adipose-Derived Stem Cells

**DOI:** 10.3390/ijms25073931

**Published:** 2024-04-01

**Authors:** Tae Gen Son, Yoojin Seo, Won-Tae Kim, Meesun Kim, Seon Jeong Choi, Si Ho Choi, Byung-Jun Sung, Jae-Seok Min, Eon Chul Han, Hyung-Sik Kim

**Affiliations:** 1Research Center, Dongnam Institute of Radiological and Medical Science, Busan 46033, Republic of Korea; kimwt33@dirams.re.kr (W.-T.K.); mskim@dirams.re.kr (M.K.); tjswjd5954@dirams.re.kr (S.J.C.); sihochoi@dirams.re.kr (S.H.C.); bj23.sung@dirams.re.kr (B.-J.S.); 2Department of Oral Biochemistry, Dental and Life Science Institute, School of Dentistry, Pusan National University, Yangsan 50612, Republic of Korea; amaicat@pusan.ac.kr; 3Department of Surgery, Dongnam Institute of Radiological and Medical Science, Busan 46033, Republic of Korea; mdoogy@dirams.re.kr (J.-S.M.); eonchulhan@dirams.re.kr (E.C.H.)

**Keywords:** three-dimensional (3D) culture, mesenchymal–epithelial transition (MET), regenerative potential, adipogenic differentiation

## Abstract

Although stem cells are a promising avenue for harnessing the potential of adipose tissue, conventional two-dimensional (2D) culture methods have limitations. This study explored the use of three-dimensional (3D) cultures to preserve the regenerative potential of adipose-derived stem cells (ADSCs) and investigated their cellular properties. Flow cytometric analysis revealed significant variations in surface marker expressions between the two culture conditions. While 2D cultures showed robust surface marker expressions, 3D cultures exhibited reduced levels of CD44, CD90.2, and CD105. Adipogenic differentiation in 3D organotypic ADSCs faced challenges, with decreased organoid size and limited activation of adipogenesis-related genes. Key adipocyte markers, such as lipoprotein lipase (LPL) and adipoQ, were undetectable in 3D-cultured ADSCs, unlike positive controls in 2D-cultured mesenchymal stem cells (MSCs). Surprisingly, 3D-cultured ADSCs underwent mesenchymal–epithelial transition (MET), evidenced by increased E-cadherin and EpCAM expression and decreased mesenchymal markers. This study highlights successful ADSC organoid formation, notable MSC phenotype changes in 3D culture, adipogenic differentiation challenges, and a distinctive shift toward an epithelial-like state. These findings offer insights into the potential applications of 3D-cultured ADSCs in regenerative medicine, emphasizing the need for further exploration of underlying molecular mechanisms.

## 1. Introduction

Once primarily recognized for its role as an energy reservoir, adipose tissue has evolved into a multifaceted entity with far-reaching implications [1]. Beyond its conventional function, adipose tissue is an intricate endocrine and secretory organ, producing adipokines that regulate essential physiological responses such as metabolic homeostasis, immune modulation, and angiogenesis [2,3,4]. The growing clinical demand for effective adipose tissue regeneration and replacement strategies underscores the need for innovative approaches to address tissue loss resulting from diverse factors such as oncological resections, traumatic injuries, and aesthetic enhancements [5,6,7,8].

Stem cells offer a promising avenue for harnessing the potential of adipose tissues. These cells are commonly utilized in two-dimensional (2D) cell culture settings, the standard approach for studying adipose-derived stem cells (ADSCs) and their properties. However, previous studies on 2D ADSCs have demonstrated some limitations. These include replicative senescence [9], cell cycle arrest [10], alterations in cell metabolism [11], and loss of differentiation potential [12]. These challenges highlight the need for more physiologically relevant approaches, such as three-dimensional (3D) ADSC cultures, to better preserve the regenerative potential and functionality of ADSCs for various applications in regenerative medicine and tissue engineering.

In recent years, the landscape of regenerative medicine has been revolutionized with the emergence of 3D organotypic cultures, which offer a platform that closely recapitulates the complex microenvironment of living tissues [13,14]. In 3D cell culture techniques, the intricate interplay between signaling molecules and cellular microenvironments is pivotal for the successful generation and maintenance of organoids [15]. Sato et al. achieved this by fostering the growth of intestinal stem cells, distinguished by the presence of leucine-rich repeat-containing G protein-coupled receptor 5 (LGR5+), in a supportive culture medium enriched with growth factors tailored to mimic the natural stem cell environment and promote tissue-specific development [16]. Wnt3a, a prominent ligand of the Wnt signaling pathway, and R-Spondin, a potent enhancer of Wnt signaling, play pivotal roles in orchestrating the fate and behavior of stem cells within 3D organoid cultures [17,18]. The interplay between these signaling molecules profoundly impacts stem cell maintenance, proliferation, and differentiation [19].

The existing literature provides a comprehensive understanding of the evolving role of adipose tissue, highlighting its complex functions beyond energy storage. However, a gap in the literature is the lack of a detailed description of the limitations of traditional two-dimensional (2D) culture methods for adipose-derived stem cells (ADSCs). Previous studies have highlighted issues such as clonal senescence, cell cycle arrest, altered cell metabolism, and loss of differentiation potential in 2D ADSC culture, but more substantial experimental results are needed for potential solutions or alternative approaches.

In the present study, we established a comprehensive framework for optimizing the 3D organotypic culture conditions for ADSCs and explored their cellular properties. Our findings reveal the full potential of ADSC-derived organoids in regenerative medicine, disease research, and drug screening.

## 2. Results

### 2.1. Establishment of 3D Organotypic ADSCs

The 3D organotypic culture of ADSCs was obtained using various concentrations of Wnt3a- and R-Spondin1-conditioned media (CM). Stem cells isolated from the adipose tissue of mice were plated in Matrigel and cultured with Wnt3a_CM 0%, R-SPO1_CM 0%, R-SPO1_CM 5%, Wnt3a_CM 25%, R-SPO1_CM 25%, Wnt3a_CM 50%, and R-SPO1_CM 10% with a control of FBS 10% and conventional 2D ADSC culture media (Figure 1A). In the control condition, the cells presented singly, whereas, in the 3D culture media condition, the cells grew as organoids, with or without Wnt3a and R-Spondin1.

The ADSCs were cultured in Matrigel, allowing 3D organotypic culture and increasing organoid size from days 3 to 11 (Figure 1B). Moreover, the 3D organotypic cultures of ADSCs were passaged multiple times (Figure 1C), confirming that the mRNA expression of Ki67, a cell proliferation marker, was well maintained and even increased at passage 13 (Figure 1D).

These results demonstrate that ADSCs are capable of long-term culture maintenance, similar to other organs, such as the small intestine and liver.

### 2.2. Validation of 3D Organotypic ADSCs on Mesenchymal Stem Cell (MSC) Characteristics

To evaluate the MSC phenotype of 2D- and 3D-cultured ADSCs, flow cytometric analysis was performed using antibodies against CD34, Sca-1, CD44, CD90.2, and CD105. Representative flow cytometry analysis results for mouse #1 ADSCs are shown in Figure 2A, and additional data for other mice (#2, #3, and #4) are shown in Appendix A (2D culture in Appendix A and 3D culture in Appendix A). Under 2D culture conditions, all surface markers, except for the negative marker CD34, were well expressed (Figure 2A). However, in 3D culture (Figure 2A), Sca-1 expression was comparable to that in the 2D culture, whereas CD44 expression was approximately half the expression level. Notably, CD90.2 and CD105 did not exhibit significant expression in 3D culture compared to 2D culture conditions. The percentage of cells positive for each antibody was quantified using fluorescence-activated cell sorting (FACS) (Figure 2B). These results indicate that 2D- and 3D-cultured ADSCs exhibit distinct differences in their MSC phenotypes.

### 2.3. Investigation of 3D Organotypic ADSC Differentiation

Adipocyte differentiation was performed to investigate the differentiation potential of 3D organotypic ADSCs. The cells were cultured in the indicated differentiation media for 14 days and cell images were visualized on the 14th day (Figure 3). The results showed that during the differentiation process of each respective cell lineage, the organoid size decreased compared to the nondifferentiation control group, indicating an overall unfavorable cellular state. In particular, on the 14th day of differentiation into adipocytes, when performing Oil Red O staining to confirm adipogenic differentiation (Figure 3B), we observed that staining did not occur effectively. Only peripheral cells of the matrix stained positive for Oil Red O, indicating adipogenic differentiation. Alternatively, we stained the cells with LipidTox, which fluorescently stains fat but cannot confirm differentiation into adipocytes (Appendix A). To confirm differentiation at the molecular level, mRNA expression of the adipocyte markers lipoprotein lipase (LPL) and AdipoQ was determined using adipocytes differentiated from 2D ADSCs as positive controls. LPL and AdipoQ mRNA expression were not detected in 3D ADSCs (Figure 3C).

### 2.4. Evaluation of Adipogenesis Activation in 3D Culture of ADSCs

Total RNA sequencing was performed to study the molecular signatures of the 2D- and 3D-cultured ADSCs. Hierarchical clustering revealed distinct gene expression patterns between the two groups (Figure 4A). A total of 1797 and 1748 genes were identified as differentially expressed, upregulated (fold change > 2, *p* < 0.05), and downregulated (fold change < 0.5, *p* < 0.05), respectively, in the 3D samples. To explore the adipogenic capacity of MSCs after 3D culture, we compared gene expression levels in 2D and 3D samples with key candidate genes reported for adipogenesis, as previously described [20]. Fourteen of the twenty genes were differentially regulated between the 2D and 3D samples (Figure 4B), demonstrating alterations in the adipogenic differentiation of 3D ADSCs. The expression of adipocyte marker genes was examined to investigate whether the 3D culture of ADSCs activates the adipogenesis pathway. Images of 2D MSCs differentiating into adipocytes (Figure 5A), 2D ADSCs (Figure 5B), and 3D ADSCs (Figure 5C) are shown. The mRNA expression levels of peroxisome proliferator-activated receptors (PPARs) (Figure 5D) and LPL (Figure 5E) were measured in 2D and 3D ADSCs, with 2D MSCs differentiating into adipocytes and serving as positive controls. Neither PPAR nor LPL mRNA expression was detected in 2D or 3D ADSCs. In contrast, the significant expression of these genes was observed in the differentiated adipocytes. These results indicate that the 3D culture of ADSCs may not be associated with changes in the expression of adipogenesis-related genes. These results suggest that the stem cell properties of ADSCs may change during 3D culture.

### 2.5. Induction of Mesenchymal–Epithelial Transition (MET) in 3D-Cultured ADSCs

We conducted a gene set enrichment analysis (GSEA) with total gene expression values. Several terms related to the proliferation category in hallmark gene sets, including ‘E2F targets’, ‘G2M checkpoint’, ‘MYC targets’, and ‘mitotic spindle’, were significantly enriched in 2D MSCs (Figure 6A). Furthermore, the gene dataset of 2D MSCs exhibited a noticeable enrichment for ‘epithelial–mesenchymal transition (EMT)’, suggesting that 3D-cultured ADSCs might display diminished mesenchymal characteristics compared to their 2D-cultured counterparts (Figure 6B). We also performed GSEA using Gene Ontology biological process (GO-BP) terms and found that 2D MSC transcriptomes were enriched in EMT. In contrast, 3D ADSCs showed a positive correlation with mesenchymal–epithelial transition (MET) (Figure 6C) and epithelial cell development (Figure 6D), indicating that our 3D culture method induced epithelial cell-like differentiation of MSCs. The expression of mesenchymal and epithelial markers was examined in 2D- and 3D-cultured ADSCs to confirm whether the 3D culture of ADSCs induces MET. The mRNA expression levels of epithelial markers E-cadherin, CK18, and EpCAM were examined to determine whether MET occurred in 3D-cultured ADSCs compared with 2D-cultured ADSCs. The results showed significantly higher E-cadherin and EpCAM mRNA expression in 3D ADSCs, while CK18 mRNA expression remained unchanged under both culture conditions (Figure 7A).

Furthermore, the expression of mesenchymal marker proteins, including α-SMA, fibronectin, vimentin, and N-cadherin, showed a significant decrease in 3D-cultured ADSCs. In particular, the expression of epithelial markers showed distinct patterns in 2D- and 3D-cultured ADSCs (Figure 7B). For example, the protein expression levels of ZO-1 and EpCAM decreased in 3D cultures, whereas that of E-cadherin increased. In contrast, there was no significant change in CK18 expression in neither 2D nor 3D ADSC cultures. Interestingly, the expression of loading control proteins beta-actin and vinculin differed between 2D- and 3D-cultured ADSCs. In 3D cultures, both proteins exhibited decreased expression in ADSCs. The expression of glyceraldehyde 3-phosphate dehydrogenase (GAPDH) was consistent in 2D- and 3D-cultured ADSCs, showing no significant difference in expression levels between the two culture conditions. All original blots were presented (Appendix A). These findings suggested that 3D culture modulates protein expression and partially promotes MET expression in ADSCs.

## 3. Discussion

The regenerative potential of stem cells has triggered a revolution in modern medicine, offering unprecedented possibilities for tissue repair and disease treatment [21]. Among these, ADSCs have emerged as a versatile cell source because of their abundance in adipose tissue and ability to differentiate into various cell lineages [22]. Although traditional two-dimensional (2D) cell culture methods provide valuable insights, they often fail to capture the complex interactions and dynamic behavior of cells within their native microenvironments. Recognizing this limitation, researchers have turned their attention to three-dimensional (3D) organotypic cultures to bridge the gap between conventional cell cultures and in vivo conditions [23,24,25]

This study aimed to establish organotypic cultures of ADSCs by harnessing the synergistic effects of conditioned media (CM) augmented with Wnt3a and R-Spondin1 to facilitate organoid formation [13,16,17,18,19]. Through systematic manipulation of the ratios of these pivotal signaling factors, the primary objective of this study was to elucidate the impact of a 3D microenvironment on ADSC growth patterns and cellular organization. However, in our 3D culture experiments, ADSCs exhibited robust organotypic culture regardless of the presence of Wnt3a and R-Spondin. We, therefore, expected this to be related to the expression of the Lgr5 gene in 3D ADSCs, and indeed, when we checked the expression of the Lgr5 gene, we found that it was absent or significantly lower in 3D ADSCs, unlike in other tissues of the liver or pancreas.

Previous studies have developed 3D cultures of ADSCs and characterized the cells. Still, the difference from our study is that they either cultured scaffold-free spheroids or planted them on hydrogels but they did not grow in an organotypic manner. For example, ADSCs spheroids are formed using the hanging-drop technique, then transferred to agarose-coated cell culture dishes [26] or cultivated on ultra-low-attachment plates [27]. A scaffold-free 3D spheroid indicated the successful generation of physiologically relevant adipose organoids with a fraction of ADSCs. In contrast, hydrogels can be scaffolds that serve as supportive structures for cell growth and allow cells to adhere and proliferate in a 3D configuration [9]. However, previous studies have only implemented temporary 3D cultures, and long-term cultures, a hallmark of organoids, have not been attempted and are probably challenging.

Throughout in vitro expansion, ADSCs undergo progressive alterations characterized by the loss of their characteristic spindle shape, multilineage differentiation potential, and self-renewal capacity [28]. Ultimately, this leads to replicative senescence, posing a notable challenge for clinical applications requiring substantial cell quantities. Hence, this study explored the potential for multiple passages of the newly developed 3D ADSCs. In addition, the sustained expression of the cell proliferation marker Ki67 [29], even after multiple passages, signifies the stability and robustness of ADSCs under these novel conditions.

The striking disparities between these two culture environments underscore the intricate interplay between the cellular microenvironment and surface marker expression, revealing potential shifts in underlying MSC characteristics [30,31]. Using flow cytometry, we investigated CD34, Sca-1, CD44, CD90.2, and CD105 expression profiles in the 2D- and 3D-cultured ADSCs. The results showed that the expression of some surface proteins was significantly reduced in the 3D cultures. Other studies that excluded differences in culture systems and examined changes in the expression of surface markers between 2D and 3D ADSC cultures have shown results different from ours. For example, Hodge et al. reported that a 2D culture system had a significantly enhanced rate of loss of stem-like CD markers (CD73, CD90, and CD105) compared to a 3D system, indicating that the 3D culture system better preserved these markers [32]. In our case, we attributed this to the different culture systems rather than a decrease in the expression of these markers as the cells were passaged [33].

Recent insights suggest that 3D culture models mimic the properties of native tissues more closely [32,34]. This investigation explored the impact of ADSC-CM derived from 2D and 3D culture systems on wound-healing cells, specifically keratinocytes (KCs) and fibroblasts (FBs) [32]. ADSC-CM from the 3D system significantly increased the metabolic, proliferative, and migratory activities of KCs and FBs, suggesting an enhanced regenerative potential. In light of recent insights, while acknowledging the limitations of our study in not directly comparing the functional disparities between our organotypic 3D culture and conventional 3D spheroid culture, it becomes evident that significant benefits emerge from the comparison between 3D spheroid culture techniques and traditional 2D culturing. The genes related to stemness, aging, telomere length, and oxidative stress were compared in 3D and 2D cultures of ADSCs [34]. We also observed distinct gene expression patterns in 2D- and 3D-cultured ADSCs. Neither PPAR nor LPL mRNA expression was detected in 2D or 3D ADSCs, whereas significant expression of these genes was observed in differentiated adipocytes. While some adipogenesis-related genes showed differential regulation, the 3D culture did not appear to activate the adipogenesis pathway. 

Interestingly, our study revealed that 2D-cultured MSCs displayed mesenchymal characteristics, whereas 3D cultures induced a shift toward an epithelial-like state (MET) in ADSCs. This shift in cell phenotype was accompanied by significant changes in the expression of various markers associated with the mesenchymal and epithelial states, demonstrating the modulatory effect of the 3D culture on cell characteristics and suggesting its potential role in promoting MET in ADSCs. Past studies investigated the therapeutic potential of stem cell-based treatments in the context of corneal injuries in mouse models [35,36]. Those studies demonstrated that human ADSC-derived epithelial-like cells, through mesenchymal–epithelial transition (MET), successfully restored the corneal epithelium in cases of persistent epithelial defects (PEDs) associated with limbal stem cell deficiency (LSCD). This study suggests that ADSCs could serve as a viable source of adult stem cells for potential autologous cell-based therapies for treating corneal surface disorders.

The limitations of this study include the lack of experiments exploring different growth factors in 3D culture media. In future research, it will be essential to explore the functional implications of the observed MET induction and assess the potential therapeutic applications of 3D-cultured ADSCs in various contexts, such as regenerative medicine and tissue engineering. Additionally, investigating the molecular mechanisms underlying this phenotypic shift in ADSCs within 3D culture systems would provide valuable insights into optimizing and controlling their differentiation for specific applications.

## 4. Materials and Methods

### 4.1. Establishment of 3D Adipose Stem Cell Culture

Twelve-week-old male BALB/c mice were obtained from Central Laboratory Animals (Seoul, Republic of Korea). The mice were euthanized by CO_2_ inhalation, and the gonadal adipose tissue was carefully dissected and separated from the surrounding tissues. The collected gonadal adipose tissue was rinsed with Hanks’ balanced salt solution (Thermo Fisher Scientific, Waltham, MA, USA) to remove blood and debris. The adipose tissue was placed in a sterile container and minced into small fragments. An enzymatic digestion was performed using collagenase type I (Sigma–Aldrich, St. Louis, MO, USA), and the tissue fragments were incubated at 37 °C for 1 h with gentle agitation to facilitate cell dissociation. The dissociated cells were passed through a sterile mesh filter (40 μm-Corning, Corning, NY, USA) to remove any remaining tissue debris and obtain a single-cell suspension. The resulting cell suspension was centrifuged at 300× *g* for 5 min, and the cell pellet was resuspended in Matrigel (Corning). The resuspended cells were seeded onto 12-well plates and incubated in various media compositions: FBS 10%, Wnt3a CM 0% + RSPO1 CM 10%, 55% RSPO1 CM, Wnt3a CM 25% + RSPO1 CM, and Wnt3a CM 50% + RSPO1 CM 10%. For passaging, 3D ADSCs were harvested using a cold harvest solution (R&D systems, CAT # 3700-100-01) on ice for 30 min to remove Matrigel. The cells were then spun down at 1200 rpm for 5 min and the supernatant was removed. The pellet was resuspended in Matrigel and seeded into 12-well plates with culture media. Cell images were obtained at the respective time points and conditions using an OLYMPUS CKX41 (Olympus Corporation, Shinjuku, Tokyo, Japan) and EVOS XL (Thermo Fisher Scientific, MA, USA). 

### 4.2. RNA Extraction and Quantitative Polymerase Chain Reaction (qPCR)

RNA extraction was performed using TRIzol reagent (Invitrogen, Waltham, MA, USA), followed by quantification using a NanoDrop 2000 spectrophotometer (Thermo Scientific) and integrity assessment with FastGene FAS-DIGI PRO (NIPPON GENETICS EUROPE GmbH, Düren, Germany). Reverse transcription utilized the PrimeScriptTM RT Reagent Kit (Takara, Shiga, Japan) to generate cDNA. Quantitative PCR was conducted on the BIO-RAD CFX96 Real-Time PCR Detection System using TOPreal™ SYBR Green qPCR UDG PreMIX (Enzynomics, Munich, Germany) with cycling conditions of 95 °C for 5 min, followed by 40 cycles of 95 °C for 30 s, 55 °C for 30 s, and 72 °C for 30 s. Melt curve analysis was performed at 55 °C for 2 s. Data were analyzed using GraphPad Prism 9 software (La Jolla, CA, USA), presenting results as fold changes or relative expression levels compared to a reference or control group.

### 4.3. Flow Cytometry Analysis for Stem Cell Confirmation

Flow cytometry was conducted to confirm the presence of stem cells. Stem cells at passage 13–14 were labeled with fluorescently labeled antibodies against CD34, Sca-1, CD44, CD90.2, and CD105, diluted at a ratio of 1:50. All antibodies used were purchased from eBioscience™ (San Diego, CA, USA). These included CD34 Isotype: Rat IgG2a kappa Isotype Control (eBR2a), FITC; CD34 Monoclonal Antibody (RAM34), FITC; CD44, CD90.2 Isotype: Rat IgG2b kappa Isotype Control (eB149/10H5), PE; CD44 Monoclonal Antibody (IM7), PE; CD90.2 (Thy-1.2) Monoclonal Antibody (30-H12), PE; CD105 Isotype: Rat IgG2a kappa Isotype Control (eBR2a), PE; CD105 (Endoglin) Monoclonal Antibody (MJ7/18), PE; Sca-1: Ly-6A/E (Sca-1) Monoclonal Antibody (D7), Alexa Fluor^®^ 488. Following incubation in the dark for 30 min at room temperature, cells were washed with PBS to remove excess antibodies. Subsequently, stained cells were resuspended in PBS for flow cytometry analysis. Analysis was performed using a Navios flow cytometer (Beckman Coulter, Brea, CA, USA) and data analyzed with Navios System Software v1.2 (Beckman Coulter). Stem cells were identified based on positive expression of specific markers.

### 4.4. Western Blot Analysis

Protein samples were extracted from cells or tissues using RIPA buffer (Thermo Fisher Scientific) supplemented with protease and phosphatase inhibitors. The protein concentration was determined using a Bradford assay kit (Bio-Rad, Hercules, CA, USA). Samples were denatured with β-mercaptoethanol (Sigma-Aldrich) and heating, then separated on 10% SDS-PAGE gels (Bio-Rad) with Dual Color Standards (Bio-Rad) for size estimation. Proteins were transferred onto nitrocellulose or PVDF membranes (Thermo Fisher Scientific) using Tris-glycine buffer with methanol (Bio-Rad). The membranes were blocked with 5% nonfat milk or BSA (BD DIFCO or Bovogen Biologicals, Keilor East, VIC, Australia) and incubated overnight with primary antibodies (1:1000 dilution). Antibodies used were α-SMA (Abcam, Cambridge, UK), fibronectin (Abcam), N-cadherin (Cell Signaling, Danvers, MA, USA), vimentin (Abcam), ZO-1 (Abcam), CK18 (Invitrogen), EpCAM (Santa Cruz Biotechnology, Dallas, TX, USA), E-cadherin (Cell Signaling), vinculin (Santa Cruz Biotechnology), β-actin (Santa Cruz Biotechnology), and GAPDH (Cell Signaling). After washing, membranes were incubated with HRP-conjugated secondary antibodies (Santa Cruz Biotechnology) for protein visualization with ECL™ Select Western Blotting (Sigma-Aldrich). Image analysis with ImageJ (NIH) quantified band intensity, and protein expression levels were determined relative to loading controls (vinculin, β-actin, or GAPDH, Santa Cruz Biotechnology).

### 4.5. Adipocyte Differentiation Assay

ADSCs were seeded at 1 × 10^4^ cells per well in 12-well plates and allowed to adhere overnight. Adipogenic induction media (STEMCELL Technologies, Vancouver, Canada) were then added and replaced every 3 days. After 14 days, cells were fixed and stained with Oil Red O (Abcam, Cambridge, UK). 

### 4.6. Library Preparation and Sequencing

Libraries were prepared from the total RNA using the NEBNext Ultra II Directional RNA-Seq Kit (NEW ENGLAND BioLabs, Inc., Hitchin, UK). The rRNA was removed using a RIBO COP rRNA depletion kit (LEXOGEN Inc., Vienna, Austria). The rRNA-depleted RNAs were used for cDNA synthesis and shearing following the manufacturer’s instructions. Indexing was performed using Illumina indexes 1–12, and the enrichment step was performed using PCR. Subsequently, the libraries were checked using an Agilent 2100 Bioanalyzer (DNA High Sensitivity Kit) to evaluate the mean fragment size. The quantification was performed using a library quantification kit and the StepOne Real-Time PCR System (Life Technologies, Inc., Carlsbad, CA, USA). High-throughput sequencing was performed by 100% paired-end sequencing using a NovaSeq 6000 (Illumina, Inc., San Diego, CA, USA).

### 4.7. Data Analysis and Gene Set Enrichment Analysis (GSEA)

The quality control of the raw sequencing data was assessed using FastQC software (ver 1.0.0); the adapter and low-quality reads (<Q20) were removed using FASTX_Trimmer (ver 0.0.13) and BBMap [37]. The trimmed reads were then mapped to the reference genome using TopHat software [38]. The gene expression levels of genes, isoforms, and lncRNAs were estimated using fragments per kilobase per million reads (FPKM) values by Cufflinks. The FPKM values were normalized based on the quantile normalization method using EdgeR in R software (ver 3.18). Data mining and graphic visualization were performed using ExDEGA (ver 2.5.0; Ebiogen Inc., Seoul, Republic of Korea). The GSEA was performed using ‘hallmark gene sets’ with ‘EMT’, ‘MET’, and ‘epithelial cell differentiation’ categories from the GO-BP gene sets, all extracted from the Molecular Signatures Database gene sets. The normalized enrichment score (NES), nominal *p*-value, and false discovery rate (FDR) q-value were used to assess the significance of enrichment.

### 4.8. Statistical and Reproducibility Analysis

Statistical analyses were conducted using GraphPad Prism 9 software. The comparisons between the two groups were performed using one-way ANOVA with Tukey’s comparison test. All error bars represent the means ± standard deviation (SD). A *p* value < 0.05 was considered a significant difference.

## 5. Conclusions

This study highlights the capacity of 3D culture environments to elicit phenotypic transformations in ADSCs, specifically driving them toward an epithelial-like state. This transition is accompanied by significant alterations in the expression profiles of surface markers and has implications for enhancing regenerative responses in cells involved in wound-healing processes. The findings of this study open avenues for investigating the clinical relevance of 3D-cultured ADSCs and delving into the molecular mechanisms underlying these changes. This offers a promising trajectory for advancing regenerative medicine and tissue engineering. 

## Figures and Tables

**Figure 1 ijms-25-03931-f001:**
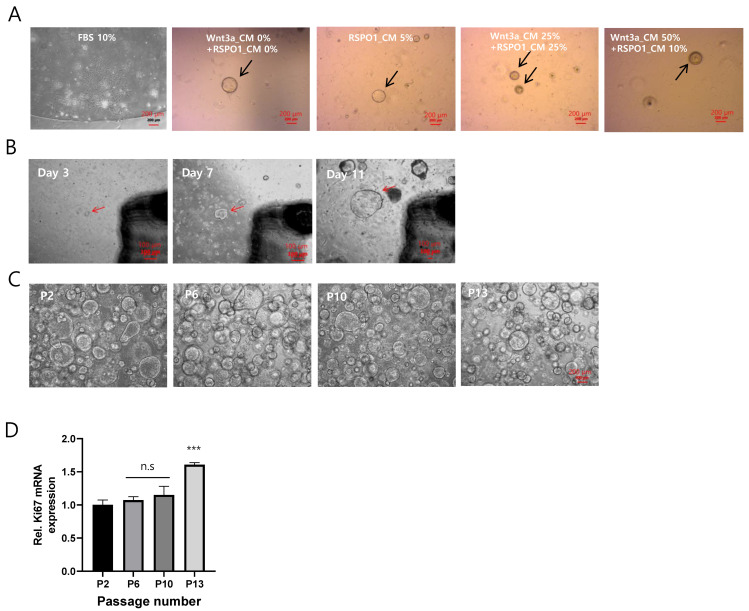
Development of three-dimensional organotypic models of adipose-derived stem cells (ADSCs). (**A**) ADSCs isolated from mouse adipose tissue were cultured in Matrigel with different combinations of Wnt3a- and R-Spondin1-conditioned media (CM): Wnt3a_CM 0% + RSPO1_CM 0%, RSPO1_CM 5%, Wnt3a_CM 25% + RSPO1_CM 25%, and Wnt3a_CM 50% + RSPO1_CM 10% and fetal bovine serum (FBS) 10% as a control. Scale bar size is 200 μm. (**B**) ADSCs exhibited organotypic growth and increased organoid size from days 3 to 11. Scale bar size is 100 μm. (**C**) Passaging of 3D organotypic cultures of ADSCs was achieved multiple times. Scale bar size is 200 μm. (**D**) The mRNA expression of Ki67, a marker for cell proliferation, was well preserved and even increased at passage 13. The values are the means ± standard deviations (*n* = 4). *** *p* < 0.001 compared with P2. Rel, relative; n.s., insignificant. The black and red arrow indicated the growth of cells in organotypic formation.

**Figure 2 ijms-25-03931-f002:**
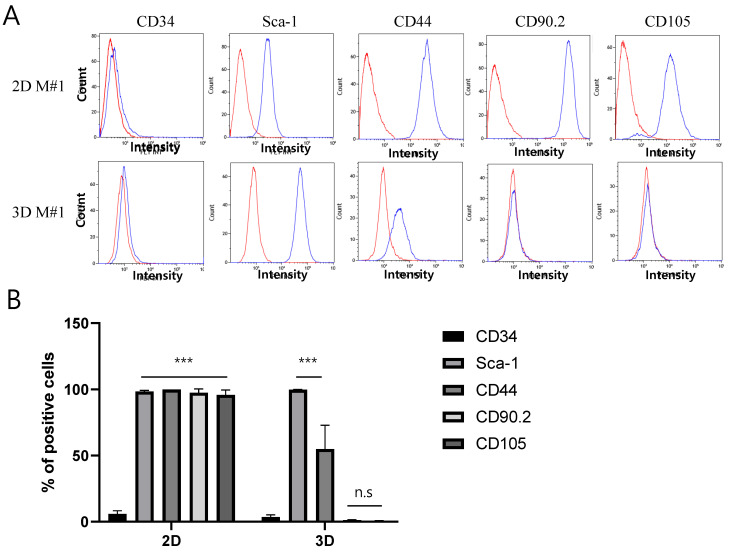
Evaluation of the mesenchymal stem cell (MSC) phenotype in 2D- and 3D-cultured ADSCs. (**A**) Representative flow cytometry analysis (FACS) mouse #1 ADSC results. The top panel represents the results of ADSCs in the 2D culture condition, while the bottom panel displays the results of ADSCs in the 3D culture condition. The isotype is represented by the red line, while the target protein is depicted by the blue line. (**B**) Quantifying the percentage of positive cells for each antibody from the FACS results. The values are the means ± standard deviations (*n* = 4). *** *p* < 0.001 compared with either 2D or 3D CD34. Rel, relative; n.s., insignificant. M#1 refers to analyzed mouse #1.

**Figure 3 ijms-25-03931-f003:**
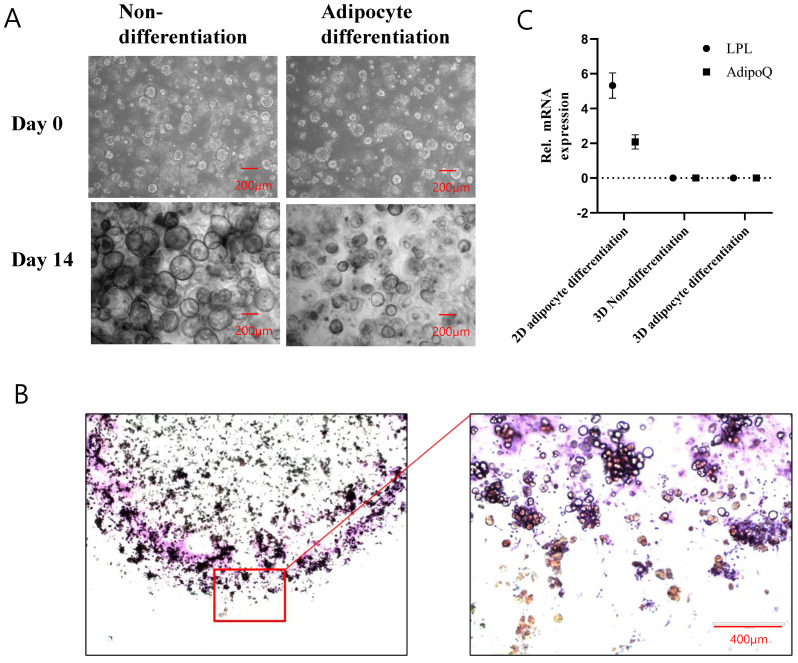
Assessment of 3D organotypic ADSC differentiation potential. (**A**) Representative images of 3D organotypic ADSCs during adipogenic differentiation on the 14th day. The decrease in organoid size relative to the nondifferentiation control group is apparent. Scale bar size is 200 μm. (**B**) Oil Red O staining performed on the 14th day of adipogenic differentiation. Positive staining for Oil Red O is observed only in peripheral cells of the matrix. Scale bar size is 400 μm. (**C**) Relative mRNA expression levels of adipocyte markers LPL and adipoQ in 3D ADSCs compared to 2D ADSCs as a positive control. The expression of LPL and adipoQ mRNA in 3D ADSCs was not detected, in contrast to their expression in 2D ADSCs. The values are the means ± standard deviations (*n* = 4).

**Figure 4 ijms-25-03931-f004:**
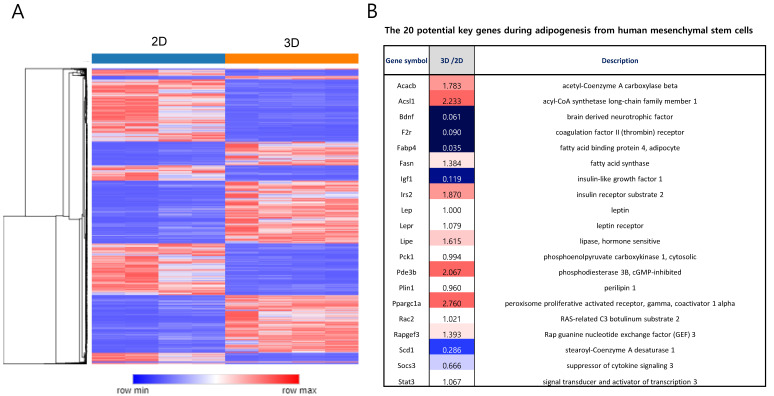
Comparative analysis of molecular signatures in 2D- and 3D-cultured ADSCs. (**A**) Hierarchical clustering analysis depicting distinctive gene expression patterns between 2D- and 3D-cultured ADSCs. (**B**) The list and relative mRNA expression levels of adipogenesis-associated key genes in 2D- and 3D-cultured ADSCs.

**Figure 5 ijms-25-03931-f005:**
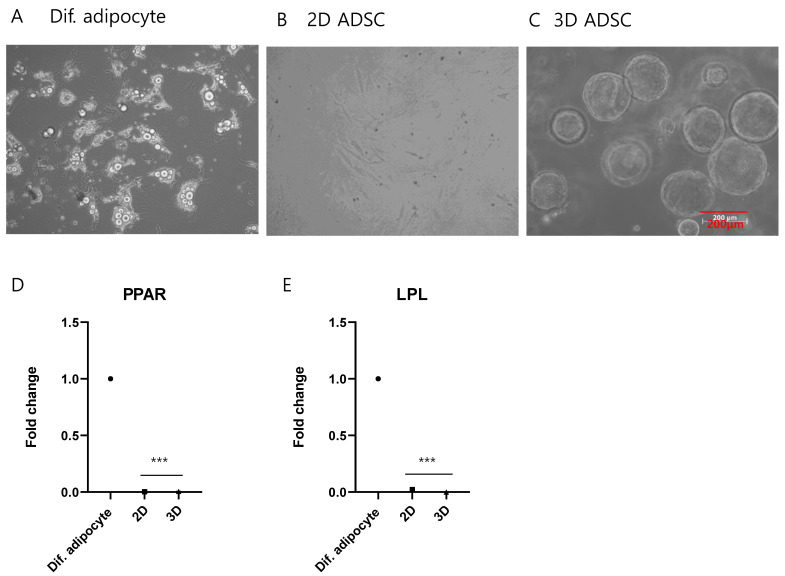
Assessment of adipogenesis pathway activation in 3D-cultured ADSCs. (**A**) Representative images illustrating the differentiation of 2D ADSCs into adipocytes. Scale bar size is 200 μm. (**B**) 2D ADSCs and (**C**) 3D ADSCs. (**D**) Analysis of mRNA expression levels of peroxisome proliferator-activated receptor (PPAR) and (**E**) lipoprotein lipase (LPL) in both 2D and 3D ADSCs, with 2D ADSCs differentiated into adipocytes serving as the positive control. The values are the means ± standard deviations (*n* = 4). *** *p* < 0.001 compared with Dif. adipocyte.

**Figure 6 ijms-25-03931-f006:**
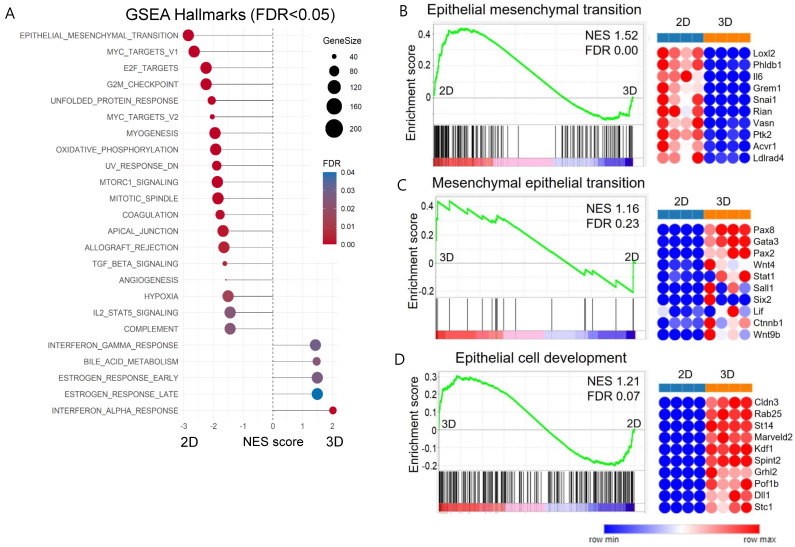
Total RNA sequencing-based functional enrichment analysis of ADSCs. (**A**) Gene set enrichment analysis (GSEA) for the hallmark gene sets using the entire gene between 2D- and 3D-cultured ADSCs. (**B**) Notable enrichment of ‘epithelial–mesenchymal transition (EMT)’ in the gene dataset of 2D ADSCs. (**C**) A positive correlation of 3D ADSCs with ‘mesenchymal–epithelial transition (MET)’. (**D**) A positive correlation of 3D ADSCs with epithelial cell development. NES, normalized enrichment score; FDR, false discovery rate.

**Figure 7 ijms-25-03931-f007:**
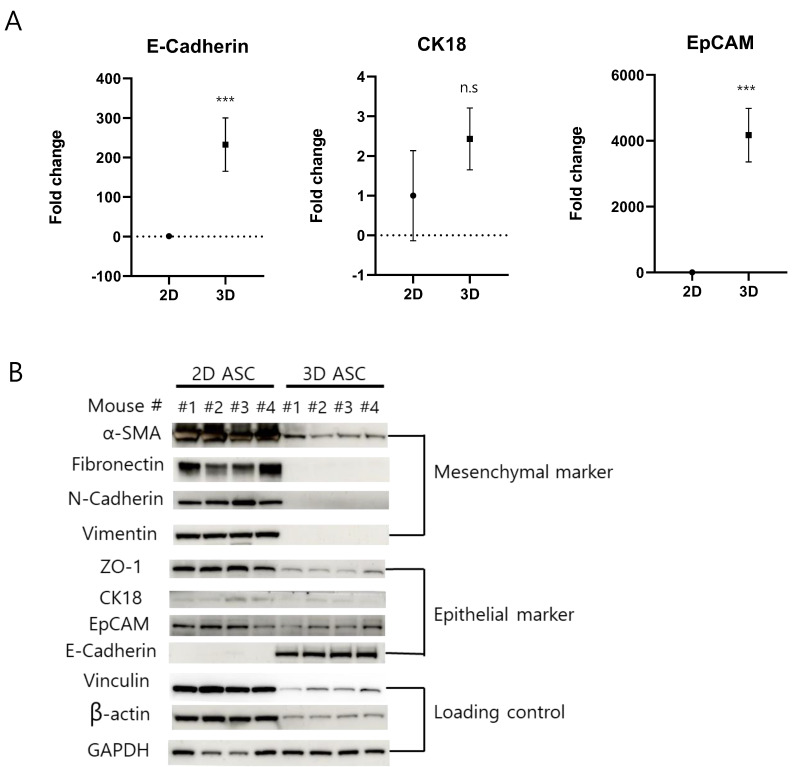
Evaluation of the expression levels of mesenchymal and epithelial markers in 2D- and 3D-cultured ADSCs. (**A**) Analysis of mRNA expression levels of epithelial markers (E-cadherin, CK18, and EpCAM) using reverse transcription polymerase chain reaction. (**B**) Assessment of the protein expression levels of mesenchymal markers (α-SMA, fibronectin, vimentin, and N-cadherin) and epithelial markers (ZO-1 and EpCAM) in 2D- and 3D-cultured ADSCs. Notably, beta-actin and vinculin showed decreased expression in 3D culture, while GAPDH remained consistent. The values are the mean ± standard deviations (*n* = 4). *** *p* < 0.001 compared with 2D; n.s., insignificant.

## Data Availability

The datasets generated for this study can be found in the repository of the GEO database (series number: GSE248060).

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
