# Peer review of "Characterization of 3D Organotypic Culture of Mouse Adipose-Derived Stem Cells"

_ijms, 2024, doi:10.3390/ijms25073931_

Round 1
Reviewer 1 Report
Comments and Suggestions for Authors
-
The study effectively highlights the differences between 2D and 3D-cultured ADSCs, emphasizing the intricate interplay between the cellular microenvironment and surface marker expression. The findings related to the modulation of ADSC characteristics, including the observed shift toward an epithelial-like state (MET), align with the evidence provided. However, the authors should acknowledge the limitations of the study, particularly regarding the lack of experiments exploring different growth factors in 3D culture media, and suggest areas for future research to address these limitations.
Major comments,
In Figure 1A, the resolution of all images in this panel is inadequate. The absence of a scale bar and the illegibility of the scale further contribute to the limitations in the image quality.
In Figure 1B &C again dealt with some poor resolution and text seems not legible.
In fig 2 panel A ordinates were not legible.
Fig 3 panel A &B, Absence of a scale bar and the illegibility of the scale further contribute to the limitations in the image quality.
Fig 3 panel C, the graphs need to be replotted.
Fig 5, Absence of a scale bar and the illegibility of the scale further contribute to the limitations in the image quality.
Fig 5 panel D, the graphs need to be replotted.
Fig 7 panel B, the blots are excessively trimmed and MW should be indicated in the blots.
Therefore, I don’t principally accept this article for publication.
Comments on the Quality of English LanguageNo major mistakes was detected.
Author Response
Dear Reviewer
Your thoughtful review of this manuscript is greatly appreciated. Enclosed, you will find comprehensive responses along with the corresponding revisions meticulously marked in the resubmitted documents.

Reviewer 2 Report
Comments and Suggestions for Authors
Son and coworkers presented a proposal for a 3D-organotypic culture of ADSCs that has a lower potential for adipogenic differentiation in the current protocol used by the authors. Several limitations are observed and some improvements can be suggested below:
A) The process of obtaining ADSC by enzymatic digestion does not only isolate ADSC from adipose tissue, therefore, the 3D structures were formed with the pool of isolated cells and not just with ADSC. This may have compromised the ADSC population present in the organoid and consequently its differentiation.
B) The antibodies that were quantified do not coincide with the antibodies that were mentioned in the methodology.
C) Lines 359-360: Please, mention the quantified antibodies, their codes, and the dilution at which they were tested.
D) Item 4.8.: Include the level of statistical significance (p-value) considered.
E) The results mention analysis up to the 13th passage, however, it was not mentioned in the methodology how these passages occurred.
F) Figures 1, 3, and 5: Scale bar of figures difficult to see and absent in some others. The legend of the figures should also mention the scale bar used.
G) Section 2.2.: When was this cytometry performed? In which passage? Please clarified.
H) Why was only adipogenic differentiation evaluated and not also chondrogenic and osteogenic differentiation? It is also missing to include in the methodology the methodological description of the differentiation assay
I) Lines 129 and 142: The authors used "at the end" of the matrix to cite differentiated cells. At the end can be substituted by "peripheral cells". Why did this result occur?
J) Some morphometric aspects such as the diameter of the organotypic structures, which were in fact similar to spheroids, could be included in the study.
K) Lines 208 to 210, and 410 to 411: Inappropriate text, please remove.
L) Discussion section: It is not clear the difference between spheroid and organoid that the study here claims to have obtained.
M) Line 279: The authors used "This study" but is not clear if the study is the present study or the study referenced in "32".
N) Line 281: What is ASC-CM?
Comments on the Quality of English LanguageSon and coworkers presented a proposal of the 3D-organotypic culture of ADSCs which shows less potential for adipogenic differentiation in the current protocol used by the authors. Several limitations are observed and need clarification.
Author Response
Dear Reviewer
Your thoughtful review of this manuscript is greatly appreciated. Enclosed, you will find comprehensive responses along with the corresponding revision meticulously marked in the resubmitted documents.

Reviewer 3 Report
Comments and Suggestions for Authors
The authors explored the use of three-dimensional (3D) cultures to maintain the regenerative capabilities of adipose-derived stem cells (ADSCs), considering the constraints associated with conventional two-dimensional (2D) culture techniques. Through a comparison between the two types of cultures, the authors observed notable differences primarily in surface markers but emphasized the occurrence of mesenchymal-epithelial transition (MET) in 3D-ADSCs. The topic holds significant value for advancing research progress. Overall, the article is well-written. The introduction covers the main points, and the objectives are accurately defined. However, some aspects of the results description and methodology could be improved.
Some considerations:
- Please remove “keyword 1” from keywords.
- Perhaps, creating a summary table outlining the advantages and limitations of the two cellular models, 2D versus 3D, could add informative value to the article. Indeed, in the introduction, the authors mention the limited information available in the literature on this topic. Therefore, it would be even more relevant to provide insights based on their observations or findings regarding this scarcity of information.
- Please explain how the different concentrations of Wnt3a- and R-Spondin1-conditioned media were defined, in material and methods section.
- Please provide the conditions for qPCR, the brand and dilutions of the different antibodies, and how many events were acquired in flow cytometry for each condition. Please carefully review the methods description and complete it. Indicate at which time point the experiments were conducted.
- I do not understand point 4.5 in the Materials and Methods section because the authors already describe RNA extraction up to point 4.2.
- There is no information concerning microscopy image acquisition and processing.
- Please add a scale bar to the figures. Additionally, the authors have only provided a description in figure 1. Please enhance the quality of the figure and/or the description of the scale.
- Please provide information regarding organoid size along the days in culture (Fig. 1b).
- The authors used the measurement of mRNA expression levels of the Ki-67 proliferative marker. Why did not the authors measure expression at the protein level (eg. Western blotting)? Evaluating Ki-67 expression at the protein level provides a direct measure of the proliferative activity of cells. Immunofluorescence or immunohistochemistry assays also can be used to detect Ki-67 expression in cells, allowing for direct visualization and localization of proliferating cells in the organoids. On the other hand, assessing Ki-67 expression at the mRNA level may not directly reflect the proliferative activity of cells, as mRNA may be present without necessarily resulting in the translation of the corresponding protein. Please elaborate on this.
- The sentence/conclusion "(...) confirming that the mRNA expression of Ki67, a cell proliferation marker, was well-maintained and even increased at passage 13 (Figure 1D)" should be reconsidered. Could the increase in the number of passages have negative repercussions on the cells' proliferation capacity?
- The figure captions should not be the same as described in the results. They should not duplicate information. Please consider making changes. In the results description, whenever possible, indicate numerical values such as mean and SD.
- In Figure 2a, please provide the information regarding the blue and red lines.
- concerning 3D-organotypic ADSC differentiation, during the 14 days, was the culture medium replaced?
- Did the authors attempt Oil Red O staining on days other than just day 14?
- Please consider changing the sentence "However, previous studies have only implemented temporary 3D cultures, and long-term cultures, a hallmark of organoids, have not been attempted and are probably impossible" regarding the word "impossible".
- Please remove “This section is not mandatory but can be added to the manuscript if the discussion is unusually long or complex.” from the conclusion.
Author Response
Dear Reviewer
Your thoughtful review of this manuscript is greatly appreciated. Enclosed, you will find comprehensive responses along with the corresponding revisions meticulosly marked in the resubmitted documents.

Round 2
Reviewer 1 Report
Comments and Suggestions for Authors
In Figure 1, I am still apprehensive about the presentation of the data. Each image appears to have been edited, but without a scale line, it's challenging to interpret their dimensions accurately. The notation of "200 micron" without a consistent scale reference adds to the confusion. There needs to be a standardized approach to image production to ensure clarity. Particularly in Panel A, the first images depicting FBS 10% differ significantly from the others in the same panel. This suggests potential variations in imaging techniques, possibly due to the use of different filters during image capture. Additionally, the absence of scale bars in any of the images further complicates their interpretation.
The flow cytometry data in Figure 2, despite being edited, appears cluttered with excessive textual information. It would benefit from a more streamlined presentation.
In Figure 3, Panel A requires scale bars with appropriate scaling in each image. Panel C lacks clarity regarding the values for 3D non-differentiated samples, necessitating a replotting to ensure visibility of these values alongside those for 3D differentiated samples. Similarly, in Panel B, the presence of scale bars is only apparent in magnified images, not in the unmagnified ones.
For Figure 5, uniformity in graph presentation is essential. Each graph should adhere to the same format for consistency.
Furthermore, all figure legends should specify the number of samples analyzed and detail the statistical tests employed to derive p-values, ensuring transparency and replicability.
Comments on the Quality of English LanguageNo major mistakes in the edited version.
Author Response
Dear Reviewer,
Please open the attached file for your review.
Sincerely,
Tae Gen Son

Reviewer 2 Report
Comments and Suggestions for Authors
Son and co-workers answered all queries satisfactorily, improving the quality of the paper.
Author Response
"Dear Reviewer,
We sincerely appreciate your feedback and acknowledgment of our efforts to address your queries satisfactorily. It is encouraging to hear that our responses have contributed to improving the quality of our paper. We have endeavored to ensure that all concerns and questions raised were thoroughly addressed, with the aim of enhancing the clarity, accuracy, and overall strength of our manuscript.
Your constructive feedback has been invaluable in guiding our revisions, and we are grateful for the opportunity to refine our work based on your insightful comments. We remain committed to upholding the highest standards of scientific rigor and clarity in our research.
Thank you once again for your thorough review and positive assessment. We look forward to the possibility of seeing our revised manuscript published in the International Journal of Molecular Science.
Sincerely,
Tae Gen Son
Reviewer 3 Report
Comments and Suggestions for Authors
I agree with the comments and responses of the authors.
Author Response

(The authors gave the same response as above.)
